# Carbon-sink potential of continuous alfalfa agriculture lowered by short-term nitrous oxide emission events

Tyler L. Anthony [1] ✉, Daphne J. Szutu[1], Joseph G. Verfaillie [1], Dennis D. Baldocchi [1] & Whendee L. Silver [1]

Alfalfa is the most widely grown forage crop worldwide and is thought to be a significant carbon sink due to high productivity, extensive root systems, and nitrogen-fixation. However, these conditions may increase nitrous oxide ($N_2O$) emissions thus lowering the climate change mitigation potential. We used a suite of long-term automated instrumentation and satellite imagery to quantify patterns and drivers of greenhouse gas fluxes in a continuous alfalfa agroecosystem in California. We show that this continuous alfalfa system was a large $N_2O$ source ($624 \pm 28$ mg $N_2O$ m$^2$ y$^{-1}$), offsetting the ecosystem carbon (carbon dioxide ($CO_2$) and methane ($CH_4$)) sink by up to 14% annually. Short-term $N_2O$ emissions events (i.e., hot moments) accounted for ≤1% of measurements but up to 57% of annual emissions. Seasonal and daily trends in rainfall and irrigation were the primary drivers of hot moments of $N_2O$ emissions. Significant coherence between satellite-derived photosynthetic activity and $N_2O$ fluxes suggested plant activity was an important driver of background emissions. Combined data show annual $N_2O$ emissions can significantly lower the carbon-sink potential of continuous alfalfa agriculture.

Alfalfa (*Medicago sativa*), a nitrogen (N) fixing species, is the most widely grown perennial forage crop worldwide and the largest crop by land area in the Western United States[1,2]. Alfalfa is traditionally used as cattle feed and growth in alfalfa land area is largely driven by increasing global feed demand for dairy and other livestock production[3]. Soil N inputs from symbiotic N fixation[4] help support plant growth but may also be a source of nitrous oxide ($N_2O$) emissions from both nitrification and denitrification[5]. From a carbon (C) accounting perspective, alfalfa has been referred to as a climate-friendly feedstock due to its soil C sequestration potential as a deep-rooting, perennial plant and reduced N fertilizer inputs[6]. However, few studies have combined carbon dioxide ($CO_2$), methane ($CH_4$), and $N_2O$ fluxes in the total net annual $CO_2$-equivalent ($CO_2$e) budgets of alfalfa agroecosystems. Continuous measurements are needed to assess the greenhouse gas emissions and net C balance of continuous alfalfa ecosystems as these

are likely to differ from other agriculture practices that incorporate alfalfa in short-term rotations[7–9].

The biogeochemical processes that drive $N_2O$ production are notorious for being temporally dynamic and characterized by hot moments of emissions, defined as short periods in time with fluxes significantly larger than the mean[5,10,11]. Thus short-term or infrequent sampling is likely to underestimate the role of hot moments in annual $N_2O$ fluxes[12]. Alfalfa typically has a high water demand and is often irrigated throughout the growing season to maintain productivity[13]. Short periods of anaerobiosis following irrigation or rainfall events combined with soil N inputs can create ideal conditions for hot moments of $N_2O$ production. Acidic conditions may exacerbate the effect as $N_2O$ reductase is inhibited at low pH[5,14]. Oxygen ($O_2$) availability is an important control on $N_2O$ production via nitrification and anaerobic denitrification[5]. Nitrous oxide production can also be

[1]Ecosystem Science Division, Department of Environmental Science, Policy and Management, University of California at Berkeley, 130 Mulford Hall, Berkeley, CA 94720, USA. ✉e-mail: t.anthony@berkeley.edu

limited by temperature, substrate C, nitrate ($NO_3^-$), or ammonium ($NH_4^+$) availability[15]. In soils, these variables are likely regulated by nonlinear asynchronous processes across temporal scales[5,16], requiring high frequency measurements to effectively characterize the controls on $N_2O$ hot moments, and determine drivers of background (i.e., non-hot moment) $N_2O$ fluxes.

The global warming potential of alfalfa agriculture is also affected by carbon dioxide ($CO_2$) and methane ($CH_4$) fluxes. Both $CO_2$ and $CH_4$ fluxes may be characterized by hot moments of soil emissions. Increased $CO_2$ respiration often occurs following soil rewetting[17-20] and these pulses can contribute a significant fraction of the annual $CO_2$ release, particularly in water-limited systems[19,21]. Heterotrophic respiration is thought to be directly regulated by substrate availability, primarily plant photosynthate[22], which can also stimulate the production and emissions of $N_2O$ and $CH_4$[23,24]. For example, root exudates are well-known labile soil C sources that can prime microbial activity and associated pulses in soil respiration[25]. Up to 20% of C fixed by photosynthesis is released by root exudation that may occur as pulsed inputs[26,27]. Changes in plant productivity within and across years may also regulate greenhouse gas fluxes through impacts on other photosynthetic inputs such as plant litter[28,29]. Alfalfa is generally a net $CH_4$ sink via microbial $CH_4$ oxidation under well-drained conditions[6,30,31], but high rainfall events and irrigation can produce anaerobic conditions that can stimulate hot moments of methanogenesis[32]. Even with the potential for periodic $CH_4$ emissions and $CO_2$ pulses, long-term eddy covariance measurements of $CO_2$ and $CH_4$ suggest that alfalfa cropping systems can be net C sinks at an ecosystem scale[6,31]. However, continuous $N_2O$ measurements are needed to determine the total $CO_2e$ of soils emissions from alfalfa.

With the increasing agricultural demand for alfalfa, continuous long-term measurements of greenhouse gas fluxes are needed to better quantify the net climate impacts of alfalfa agroecosystems. It is also critical to determine the drivers of greenhouse gas emissions to better manage alfalfa for emissions reduction. We used a combined suite of automated flux chambers, continuous environmental sensing, eddy covariance, and satellite imagery of photosynthetic activity to determine patterns and associated controls of $CO_2$, $N_2O$, and $CH_4$ fluxes over four complete years in irrigated alfalfa. We used continuous cavity ring-down spectroscopy (CRDS) and automated chambers to collect over 103,000 individual $N_2O$, $CH_4$, and $CO_2$ flux measurements which were coupled with soil $O_2$, moisture, and temperature sensors installed across the soil profile and a year-long intensive weekly sampling campaign for soil gas ($CO_2$, $N_2O$, and $CH_4$), mineral N, and soil pH. We tested the hypothesis that the combination of mineral N availability and low redox conditions are the primary drivers of hot moments of $N_2O$ emissions and that hot moments offset a significant portion of the net $CO_2e$ sink. We predicted that low redox conditions would occur during irrigation and high rainfall events, particularly during warm periods as the solubility of $O_2$ decreases with increasing temperature[33]. We also hypothesized that background patterns in $N_2O$ emissions would follow patterns in plant activity indicative of the potential impact of plants on C or substrate availability.

## Results and discussion
### Annual soil $N_2O$ budgets and ecosystem $CO_2e$ balance
Annual mean $N_2O$ fluxes were $624.4 \pm 26.8$ mg $N_2O$ m$^{-2}$ yr$^{-1}$ (Table 1, range: $247.0 \pm 5.7$ to $901.9 \pm 74.5$ mg $N_2O$ m$^{-2}$ yr$^{-1}$) and were similar to or greater than other $N_2O$ flux estimates from alfalfa systems[7,34-37]. However, few studies report flux measurements from irrigated, continuous alfalfa monocultures[36,37], which make up the majority of alfalfa ecosystems in the Western United States[2,7,13,38]. Annual soil $N_2O$ emissions were highest in site years 2 and 3 (Table 1, $p < 0.001$) and lowest in site year 4 ($p < 0.001$). The use of N-fixing crops as a means to reduce N fertilizer inputs to agroecosystems is expected to decrease overall $N_2O$ agricultural emissions[39]. However, the mean $N_2O$ fluxes observed here

($4.0 \pm 0.2$ kg N-$N_2O$ ha$^{-1}$ y$^{-1}$) were equal to or higher than rates from fertilized agricultural ecosystems[40,41]. This suggests net $N_2O$ emissions from irrigated alfalfa may not always be reduced relative to other agricultural ecosystems receiving inorganic N inputs, particularly on relatively C-rich soils.

Soil $N_2O$ fluxes reduced the annual net $CO_2e$ sink (sum of eddy covariance NEE and chamber $N_2O$ and $CH_4$; Table 1) by up to 14% (mean: $8 \pm 0.4\%$). The ecosystem was a consistent net $CO_2e$ sink (mean: $-450.4 \pm 121.9$ g $CO_2e$ m$^{-2}$ y$^{-1}$) when estimated using eddy covariance NEE observations and chamber observations of $N_2O$ and $CH_4$ (Table 1). Annual global warming potential (GWP) values were significantly greater in years 3 and 4 than years 1 and 2, driven by significant increases in net ecosystem exchange (NEE) and lower $N_2O$ fluxes in year 4 (Table 1). Annual $CH_4$ fluxes were a consistent $CO_2e$ sink (mean: $-1.5 \pm 0.1$ g $CO_2e$ m$^{-2}$ y$^{-1}$) but were always less than 0.5% of the annual net GWP.

### The importance of $N_2O$ hot moments
Inter- and intra-annual variability in $N_2O$ fluxes were largely driven by differences in the magnitude and frequency of hot moments of $N_2O$ production. Hot moments represented only 0.2 to 1.1% of annual $N_2O$ measurements but contributed up to 57% (mean: $44.4 \pm 6.3\%$) of total $N_2O$ emissions (Table 2), highlighting the importance of continuous measurements for capturing high emission events and that continuous background fluxes (i.e., lower than hot moments) still represent a significant portion of the annual budget. The magnitude of hot moments decreased with stand age, and the contribution of hot moments to the annual flux also decreased over time (Fig. 1c, Table 2, $p < 0.001$). The decrease in the magnitude of hot moments of $N_2O$ emissions over time may be partially explained by increased alfalfa taproot development with stand age. Nitrous oxide fluxes are generally expected to increase with alfalfa stand age[36], driven by increasing organic matter and N inputs from more developed root systems. However, irrigation frequency is likely to decrease in more established stands or in systems supported by subsurface irrigation or a shallow water table[42-44], which could lower $N_2O$ fluxes[45]. A well-developed taproot system can maintain access to a deep-water table to support plant water demands, reducing drought stress[46,47] and the need for irrigation events that stimulate hot moments of $N_2O$ emissions. The decreased contribution and magnitude of $N_2O$ hot moments did not consistently correspond to decreases in annual emissions (Table 2). This may be due to increases in $N_2O$ emissions associated with the accumulation and mineralization of residual alfalfa-derived organic matter[36]. Here we found that reduced irrigation frequency drove the observed decreases in hot moment emissions with stand age. However, these emissions reductions were partially offset by background (i.e., lower than hot moment) increases in $N_2O$ production, which could have been derived from greater soil C and N availability.

### Drivers of soil $N_2O$ emissions
Acidic soil conditions were maintained throughout the year (Fig. 2c), creating a favorable pH environment for incomplete denitrification following decreases in soil $O_2$ availability[48]. These soils were relatively C-rich (5% soil C from 0–30 cm)[49], which may have also contributed to the higher observed $N_2O$ emissions here[50], but newly mineralized alfalfa roots and shoots were likely an important soil $NO_3^-$ source[51] and substrate for denitrification. We found that hot moments of $N_2O$ production occurred following rapid increases in moisture and decreases in soil $O_2$ in warm surface soils; lower soil temperatures in winter appeared to limit hot moments of $N_2O$ emission following rain events (Fig. 2). Summer hourly mean $N_2O$ fluxes peaked in late afternoon (Fig. 3c, $p = 0.06$), within hours after the onset of irrigation events. However, short periods of irrigation did not always correspond to increased soil moisture at depths below 10 cm (Fig. 2). This could

**Table 1 | Annual greenhouse gas emissions**

| Year | N$_2$O flux (mg N$_2$O m$^{-2}$ y$^{-1}$) | N$_2$O GWP (g CO$_2$e m$^{-2}$ y$^{-1}$) | CH$_4$ flux (mg CH$_4$ m$^{-2}$ y$^{-1}$) | CH$_4$ GWP (g CO$_2$e m$^{-2}$ y$^{-1}$) | Chamber CO$_2$ flux (g CO$_2$ m$^{-2}$ y$^{-1}$) | NEE (g CO$_2$ m$^{-2}$ y$^{-1}$) | Total CO$_2$e (g CO$_2$ m$^{-2}$ y$^{-1}$) | Eddy R$_{eco}$ (g CO$_2$ m$^{-2}$ y$^{-1}$) | Eddy GPP (g CO$_2$ m$^{-2}$ y$^{-1}$) |
|---|---|---|---|---|---|---|---|---|---|
| 1 (2017–2018) | 610.5 ± 68.1 a | 181.9 ± 20.3 a | −44.0 ± 2.2 ab | −1.2 ± 0.1 ab | 5869.5 ± 31.4 a | −1757 ± 85 a | −1576.3 ± 105.4 a | 6485 ± 25 a | 8242 ± 96 a |
| 2 (2018–2019) | 901.9 ± 74.5 b | 268.8 ± 22.2 b | −31.6 ± 2.5 a | −0.9 ± 0.1 a | 4135.0 ± 25.4 d | −1989 ± 86 a | −1721.1 ± 108.3 a | 6141 ± 22 b | 8129 ± 96 a |
| 3 (2019–2020) | 777.1 ± 52.0 ab | 231.6 ± 15.5 ab | −60.6 ± 2.8 b | −1.7 ± 0.1 b | 5217.3 ± 23.9 b | −2942 ± 101 b | −2712.1 ± 116.6 b | 6513 ± 27 a | 9455 ± 113 b |
| 4 (2020–2021) | 263.6. ± 5.6 c | 78.6 ± 1.7 c | −78.2 ± 8.8 c | −2.2 ± 0.2 c | 4565.2 ± 26.5 c | −2632 ± 93 b | −2555.6 ± 94.9 b | 6521 ± 24 a | 9153 ± 103 b |
| All | 624.4 ± 27.8 | 186.1 ± 8.3 | −53.5 ± 2.5 | −1.5 ± 0.1 | 4925.9 ± 13.5 | −2330 ± 46 | −2115.4 ± 54.4 | 6451 ± 12 | 8745 ± 51 |

Mean (± standard error) annual chamber nitrous oxide (N$_2$O), methane (CH$_4$), and carbon dioxide (CO$_2$) fluxes, 100-year global warming potential (GWP) of N$_2$O and CH$_4$ in CO$_2$-equivalence (CO$_2$e), eddy covariance annual mean net ecosystem exchange (NEE), and annual field-scale CO$_2$e emissions (combination of chamber N$_2$O and CH$_4$ fluxes and eddy covariance NEE) by site year. Net ecosystem exchange (NEE) was derived from ecosystem respiration (R$_{eco}$) and gross primary productivity (GPP) eddy-covariance measurements. Letters denote statistically significant differences among annual values ($p < 0.01$) with statistical results reported from one-way repeated measures ANOVAs.

**Table 2 | N$_2$O fluxes**

| Year | Annual mean (mg N$_2$O m$^{-2}$ y$^{-1}$) | Flux (n) | Hot moment flux (n) | Hot moment mean (mg N$_2$O m$^{-2}$ d$^{-1}$) | Hot moments removed mean (mg N$_2$O m$^{-2}$ y$^{-1}$) | Hot moments % of total flux |
|---|---|---|---|---|---|---|
| 1 (2017–2018) | 610.5 ± 68.1 | 25,252 | 48 | 496.1 ± 66.8 | 263.4 ± 9.9 | 56.8% |
| 2 (2018–2019) | 901.9 ± 74.5 | 25,169 | 74 | 456.6 ± 43.0 | 402.4 ± 13.7 | 55.3% |
| 3 (2019–2020) | 777.1 ± 52.0 | 26,261 | 55 | 363.1 ± 46.2 | 485.7 ± 10.4 | 37.5% |
| 4 (2020–2021) | 263.6 ± 5.6 | 25,336 | 273 | 19.8 ± 0.7 | 180.39 ± 2.6 | 31.6% |
| All years | 624.4 ± 27.8 | 103,013 | 201 | 401.1 ± 26.9 | 346.9 ± 4.7 | 44.4% |

Mean (± standard error) annual nitrous oxide (N$_2$O) fluxes by site year, number of measurements, number of outlier measurements, outlier mean (± standard error) N$_2$O fluxes, mean N$_2$O fluxes (± standard error) without hot moments included, and contribution of hot moments to total mean flux. Hot moments were calculated separately for each year and in aggregate for the total dataset (All years).

indicate that the majority of N$_2$O production during hot moments occurred near the soil surface.

Soil N$_2$O fluxes were significantly greater in the summer and lowest in the fall (Fig. 3c, $p < 0.001$). Seasonal and diel trends in soil N$_2$O fluxes further emphasized the importance of soil moisture changes from irrigation and rainfall events. Mean hourly N$_2$O fluxes during summer periods, when most irrigation events occurred, were consistently greater than any other period (Fig. 3c, $p < 0.001$). Overall, daily mean N$_2$O fluxes were positively correlated with weekly soil atmosphere N$_2$O concentrations across depths, suggesting N$_2$O production across the soil profile contributed to background soil N$_2$O emissions (Supplementary Information, Fig. S1, 10 cm R$^2$ = 0.60, $p < 0.001$, 30 cm R$^2$ = 0.53, $p < 0.001$, 50 cm R$^2$ = 0.45, $p < 0.001$). Temporal patterns in soil moisture, soil temperature, and bulk soil O$_2$ concentrations covaried across all depths and were significantly related to patterns in net N$_2$O fluxes on a daily timescale (Fig. S2, $p < 0.05$). Primary interactions between N$_2$O and moisture, temperature, and O$_2$ suggested that changes in N$_2$O fluxes were generally in phase but lagged changes in these variables at daily and weekly timescales (Supplementary Information, Fig. S2, $p < 0.05$). At weekly and monthly timescales, N$_2$O fluxes were predominantly associated with soil temperature and moisture at 10 and 30 cm depths (Supplementary Information, Fig. S2, $p < 0.05$). Wavelet coherence analyses suggested that short-term, hot moments of N$_2$O emissions were stimulated by changes in moisture and O$_2$ concentrations in surface soils, as well as sustained acidic soil conditions. Acidic soil pH (Fig. 3) and lagged responses of temperature and moisture were the predominant controls at longer timescales (Supplementary Information, Fig. S2, $p < 0.05$).

Continuous low-magnitude N$_2$O production was an increasing fraction of total N$_2$O emissions over time (Table 2). In contrast to hot moments, consistent low magnitude N$_2$O fluxes were regulated by plant activity, soil moisture, and soil temperature throughout the soil profile. Increases in background (low magnitude) N$_2$O emissions were positively correlated with periods of high gross primary productivity (GPP), measured with satellite observations of near-infrared

reflectance of vegetation (NIRv, Supplementary Information, Fig. S3, $p < 0.05$)[28]. Alfalfa releases a small proportion of its symbiotically-fixed N as NH$_4^+$ to the soil[52–54], and decreases in photosynthate supply to root nodules and exudates following shoot harvest may also limit C substrate availability to nitrifiers and denitrifiers[55]. While no relationships were observed between soil NO$_3^-$ or NH$_4^+$ and N$_2$O emissions during our weekly sampling campaign (Fig. 2b, Supplementary Information), observed coherence at a daily timescale between NIRv and N$_2$O suggested plant-derived C or NH$_4^+$ availability may regulate low magnitude N$_2$O emissions. Plants likely shifted C and N allocation to new plant growth immediately after cutting, leading to lower soil N$_2$O emissions. Emissions increased over the growing season, possibly due to greater root exudation as aboveground plant biomass re-established.

**Soil CH$_4$ emissions**
Soils were a small consistent net sink of CH$_4$, accounting for only 0.06% of the total net C based-CO$_2$e uptake over the four year period. Annual mean soil CH$_4$ fluxes were −53.5 ± 2.5 mg CH$_4$ m$^{-2}$ y$^{-1}$ (Table 1, range: −78.2 ± 8.8 to −31.6 ± 2.5 mg CH$_4$ m$^{-2}$ yr$^{-1}$). The net CH$_4$ sink was significantly greater in site year 4 than all other years (Table 1, $p < 0.001$). Sinks measured here were larger than others alfalfa ecosystem estimates[10,31,56], likely from the lower detection limit of the CRDS and automated chambers. In contrast to expectations, decreases in bulk soil O$_2$ concentrations did not appear to drive significant increases in net CH$_4$ production or decreases in the CH$_4$ sink (Fig. 1). Extended periods of soil anaerobiosis may be required to stimulate net CH$_4$ production[12,57], and this was not observed during the four year measurement period. We did observe a substantial increase in soil CH$_4$ concentrations (but not surface fluxes) shortly following the largest decrease in soil O$_2$ concentrations in March and April 2019 (Fig. 1, Supplementary Information, Fig. S8). Elevated soil moisture may have limited gas diffusion. Slower diffusion together with methanotrophic consumption near the soil surface likely regulated net soil CH$_4$ efflux during this period[58,59]. We also observed significant variability in hourly mean diel CH$_4$ fluxes (Fig. 3d), but this

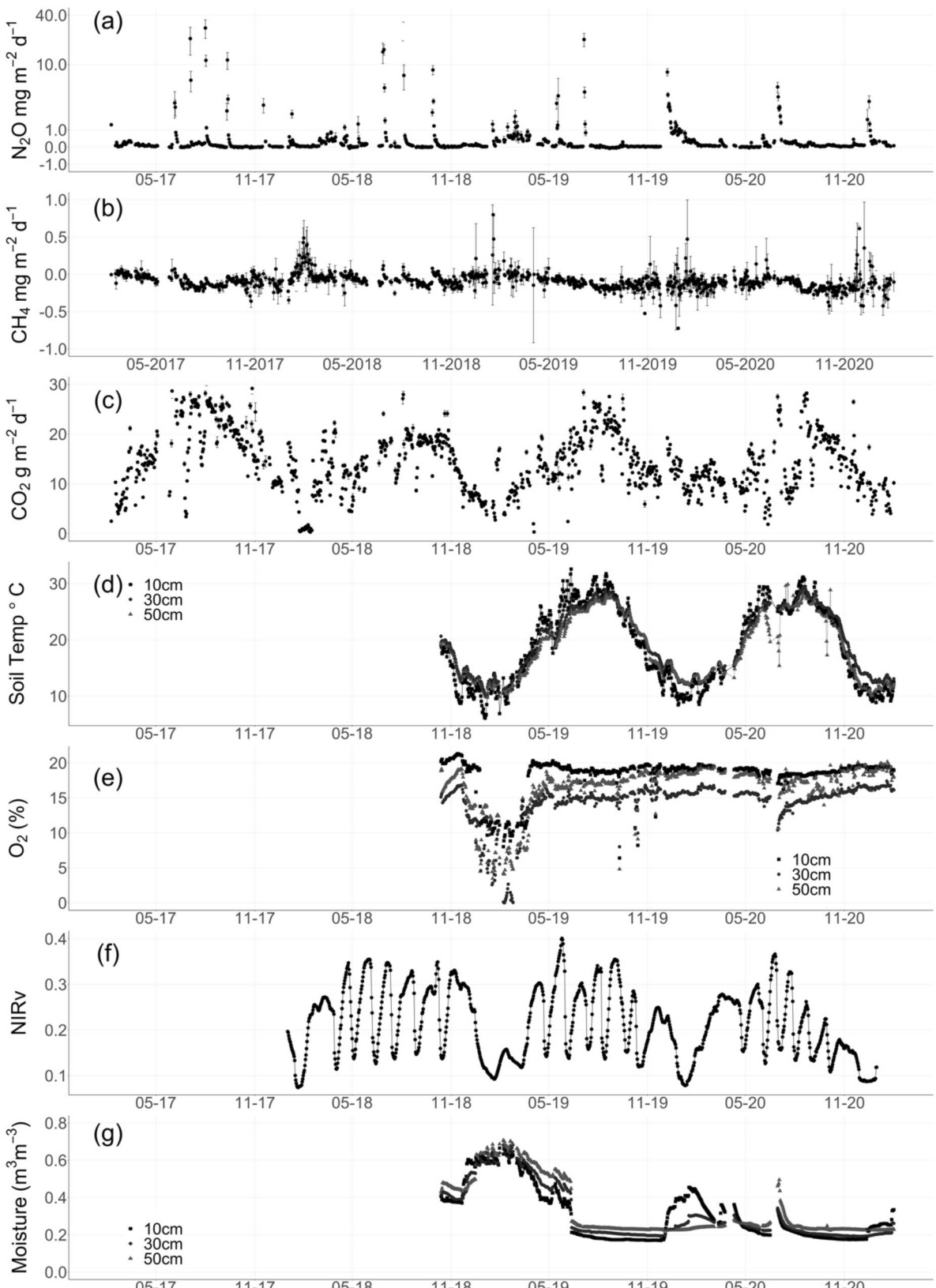

**Fig. 1 | Greenhouse gas fluxes, soil sensing, and satellite imagery.** Daily mean (± standard error) (**a**) carbon dioxide (g $CO_2$ $m^{-2}$ $d^{-1}$), **b** methane (mg $CH_4$ $m^{-2}$ $d^{-1}$), and **c** nitrous oxide (mg $N_2O$ $m^{-2}$ $d^{-1}$) fluxes ($n$ = approximately 80 per day, with a total of 108,638, 103,013, and 102,997 flux measurements of $CO_2$, $N_2O$, and $CH_4$, respectively). Black circles represent mean daily flux measurements. Daily mean (± standard error) (**d**) soil temperature (°C), **e** soil oxygen ($O_2$), **f** daily near-infrared reflectance of vegetation (NIRv), and (**g**) volumetric soil moisture ($m^3$ $m^{-3}$) over the soil sensor measurement period and available daily satellite imagery ($n$ = 96 measurements per day except for NIRv). For (**d**) soil temperature, (**e**) $O_2$, and (**g**) moisture, depth values are labeled as squares (10 cm), circles (30 cm), and triangles (50 cm).

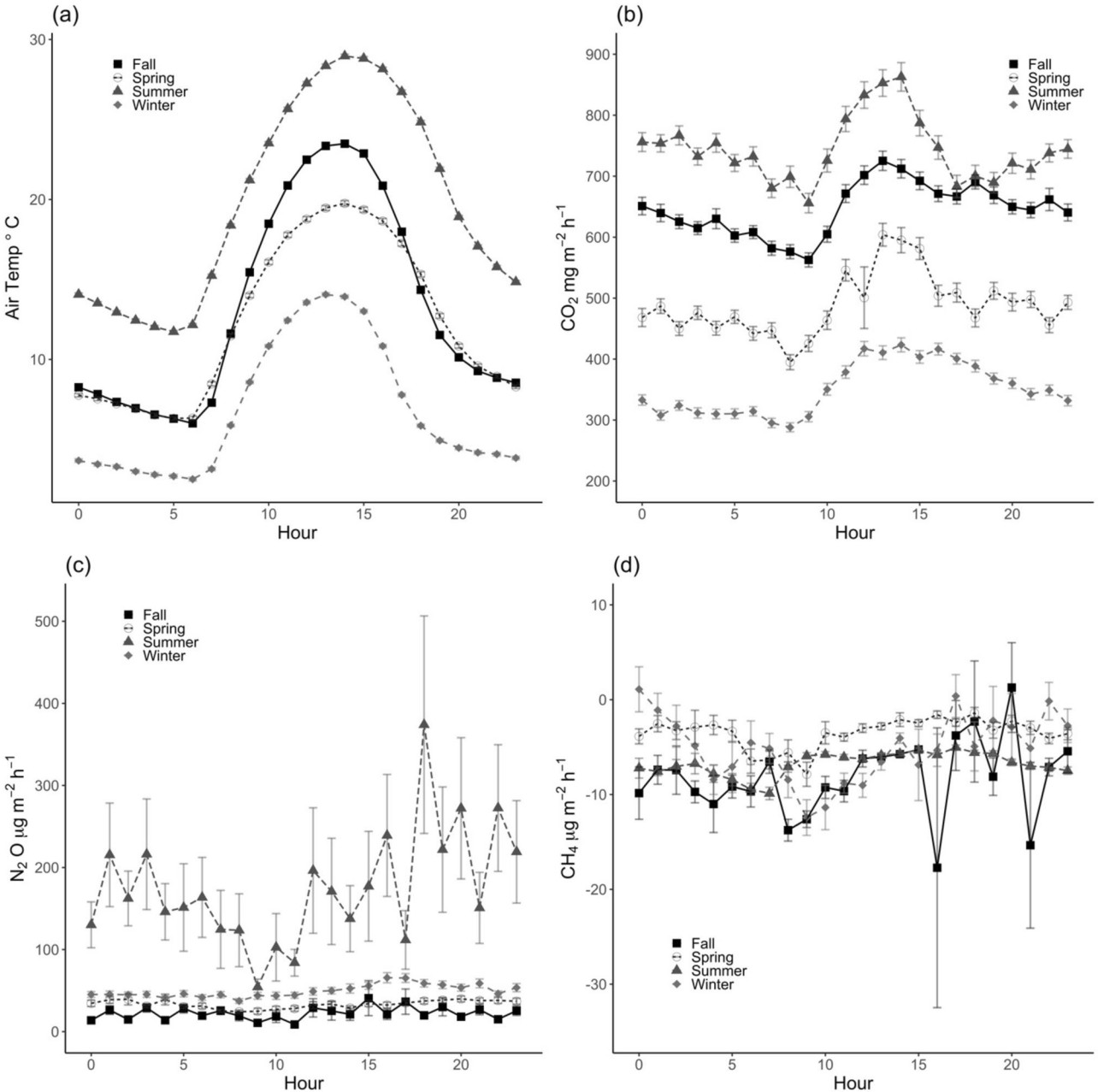

**Fig. 2 | Diel greenhouse gas fluxes.** Hourly mean (± standard error) (a) air temperature (℃), (**b**) carbon dioxide ($CO_2$) fluxes (mg $CO_2$ m$^{-2}$ h$^{-1}$), (**c**) methane ($CH_4$) fluxes (µg $CH_4$ m$^{-2}$ h$^{-1}$), and (**d**) nitrous oxide ($N_2O$) fluxes (µg $N_2O$ m$^{-2}$ h$^{-1}$), grouped by season (Fall = squares, Spring = open circles, Summer = triangles, and Winter = diamonds) over the entire measurement period (Fall: $n \geq 1220$ measurements per hour, Spring: $n \geq 848$ measurements per hour, Summer: $n \geq 956$ measurements per hour, Winter: $n \geq 1060$ measurements per hour).

variability was not significantly correlated with any measured soil characteristics.

Methane fluxes varied in response to temperature across depths and timescales and temperature was the strongest control on net $CH_4$ consumption (Supplementary Information, Fig. S4, $p < 0.05$). Decreases in soil moisture stimulated net $CH_4$ consumption (Supplementary Information, Fig. S4, $p < 0.05$), likely due to increased diffusivity[60,61] and $O_2$ availability. Periods of $CH_4$ uptake were highest in the late summer, occurring when soils were the driest throughout the soil profile (Fig. 1a, c). Lower soil moisture across the soil profile also generally corresponded to higher rates of $CH_4$ uptake and lower overall soil moisture increased $CH_4$ uptake with stand age, except for site year 2 (Table 1). Sustained $CH_4$ consumption combined with observed trends in N cycling suggest that $CH_4$ oxidation by nitrifiers or nitrification by methanotrophs[62–65] could be regulating non-$CO_2$ greenhouse gas production and consumption under oxic conditions.

## Agroecosystem $CO_2$ balance

Soil $CO_2$ emissions were greater than other alfalfa ecosystems[6], likely driven by a combination of high plant productivity, relatively high soil C content[49], and warm temperatures throughout the growing season. Chamber $CO_2$ fluxes, which here represent combined soil and root respiration, averaged $4925.9 \pm 13.5$ g $CO_2$ m$^{-2}$ yr$^{-1}$ and were lower than ecosystem respiration ($R_{eco}$) estimates ($6451 \pm 12$ g $CO_2$ m$^{-2}$ yr$^{-1}$) from the nearby eddy covariance observations (Table 1). Soil $CO_2$ fluxes closely followed seasonal patterns in soil temperature, with similar trends in soil temperature observed across depths (Fig. 1a, b). Surprisingly, soil $CO_2$ fluxes did not vary significantly with NIRv on

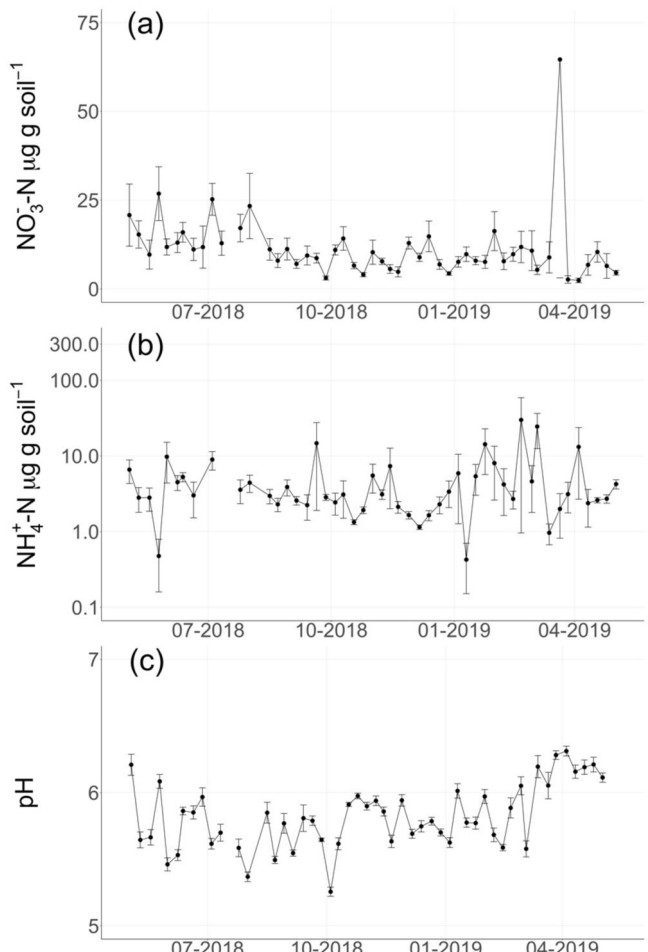

**Fig. 3 | Weekly soil nitrogen and pH.** Weekly mean (± standard error) (**a**) soil nitrate (µg $NO_3^-$-N g soil$^{-1}$), **b** soil ammonium (µg $NH_4^+$-N g soil$^{-1}$), and **c** soil pH (n = 10 per week for 52 weeks). Manual soil measurements (0–10 cm depth) were conducted weekly from May 2018–May 2019.

### Table 3 | Moisture, rainfall, and NIRv

| Year | Mean 0-50 cm Soil Moisture (%) | Rainfall (mm y$^{-1}$) | NIRv |
|---|---|---|---|
| 1 (2017-2018) | – | 444 a | – |
| 2 (2018-2019) | 47.8 ± 7.8 b* | 356 ab | 0.23 ± 0.01 a |
| 3 (2019-2020) | 35.3 ± 17.1 a | 447 a | 0.21 ± 0.01 b |
| 4 (2020-2021) | 24.3 ± 4.3 c | 176 b | 0.19 ± 0.01 c |
| All | 31.5 ± 14 | 331 ± 64 | 0.21 ± 0.01 |

Annual mean (± standard error) 0–50 cm soil moisture (n = 96 measurements per day for 763 total days), annual rainfall (mm y$^{-1}$), and mean (± standard error) annual near-infrared reflectance of vegetation (NIRv) by site year (e.g., January 27 to January 26, n = 365 per year).
*Site year 2 soil moisture values include 5 out of 12 months. Letters denote significant differences between annual values (p < 0.01) with statistical results reported from one-way repeated measures ANOVAs.

within 5 to 7 days (Fig. 1a). Periods of increased plant growth rates following harvests likely resulted in a shift in C allocation from below- to aboveground[68] and highlights the importance of substrate limitation on NEE[29]. If aboveground regrowth increases plant nutrient demands, it could induce a lagged response in belowground respiration driven by subsequent reallocation of photosynthate belowground followed by enhanced soil nutrient mining by microbial communities[69]. The observed lagged relationships between NIRv and ecosystem greenhouse gas fluxes may also represent delays between photosynthetic $CO_2$ uptake and root C exudation processes. Short-term increases in soil moisture content and associated decreases in $O_2$ availability throughout the year were also important controls on soil respiration (Supplementary Information, Fig. S5, p < 0.05). Soil temperature across depths was significantly associated with respiration rates across timescales.

This combination of automated chambers, eddy covariance, soil sensing, and satellite imagery used here provided a comprehensive dataset of multi-year, annual, ecosystem-scale fluxes from a continuous alfalfa agroecosystem. We were able to determine the importance of both short-term hot moments and background emissions on total greenhouse gas budgets and explore scale-emergent drivers of $N_2O$ emissions. We found that $N_2O$ emissions reduced the net $CO_2$e sink at the ecosystem-scale by up to 14% annually and offset 70% of the ecosystem C sink after accounting for harvest biomass removal (post-harvest fate of harvested alfalfa was not included in this calculation). As hypothesized, this was predominantly driven by rare hot moments of soil $N_2O$ emissions supplied by elevated soil $NO_3^-$ pools and acidic soil conditions and stimulated by irrigation and rainfall events. Hot moments were ≤1% of measurements but averaged 44.4 ± 6.3% of annual $N_2O$ fluxes. Additionally, background fluxes were likely driven by sustained substrate availability that varied with moisture, temperature, and NIRv, a possible index of plant inputs to soil. Lagged relationships between NIRv, $CO_2$, and $N_2O$ fluxes suggested that plant inputs were likely an important driver of soil $CO_2$ fluxes and background $N_2O$ emissions. Our results show that $N_2O$ emissions likely significantly lower the field-scale C sink potential of this globally important crop. Hot moments of $N_2O$ emissions, typically underestimated with traditional measurement approaches, played an outsized role in annual ecosystem-scale greenhouse gas budgets, highlighting the importance of continuous measurement for accurate ecosystem-scale greenhouse gas accounting.

## Methods
### Site info
The study was conducted in the Sacramento-San Joaquin Delta region of California, USA (38.11°N, 121.5°W). The site was in conventional perennial alfalfa (>5 years) that was periodically flood-irrigated during the growing season. The site was located on highly degraded

a daily scale (Supplementary Information, Fig. S3, p < 0.05). Soil $CO_2$ fluxes and NIRv covaried on weekly, monthly, and annual timescales highlighting the importance of plant harvesting and phenology in regulating soil respiration[28].

To quantify for C removed from the field, we used mean annual yields of 595 ± 137 g C m$^{-2}$ y$^{-1}$ or 2,072 ± 502 g $CO_2$ m$^{-2}$ y$^{-1}$ [31]. This is equivalent to 89% of NEE, with the remaining C (258.3 g $CO_2$ m$^{-2}$ y$^{-1}$ or 70.4 $CO_2$ m$^{-2}$ y$^{-1}$) assumed to be stored as belowground biomass or soil C. With this number, the $CO_2$e of $N_2O$ emissions would then offset 70% of the net $CO_2$ sink. The fate of harvested C was not considered in this study, which examined only ecosystem-scale fluxes. However, it should be noted that if conducting a life cycle analysis, harvested alfalfa is typically used as dairy or cattle feed, where alfalfa C is converted to a combination of both $CO_2$ and $CH_4$[66].

We observed significant differences in NIRv following alfalfa cutting events (Fig. 1c) and mean annual NIRv decreased significantly across the measurement period (Table 3, p < 0.01). Lags observed between NIRv and soil $CO_2$ fluxes may represent a delayed response to photosynthate availability as plants likely reallocate new photosynthate to aboveground biomass production following harvest events[67]. Alfalfa in this region can be harvested up to seven times per year, where the majority of aboveground plant biomass is removed[28,31]. Cuttings corresponded to significant reductions in mean daily soil respiration values, although soil respiration values typically recovered

peatland soils that have lost a significant proportion of their initial organic matter[49]. Alfalfa and corn are the dominant agricultural land uses in the region, with alfalfa representing 20% of agricultural land area (32,000 ha) in the Sacramento-San Joaquin Delta[70] and the largest crop by area in California (405,000 ha)[71]. Nearly 100% of alfalfa in California is irrigated, with flood irrigation being the most common practice[71]. The site had a Mediterranean climate with hot dry summers and cool wet winters. The region's historical mean annual temperature was $15.1 \pm 6.3\,°C$ and a mean annual rainfall averaging $326 \pm 4\,mm$[23]. Site year (January 27 - January 26) rainfall data was collected from a nearby (<1 km) Ameriflux site[72]. Near-infrared reflectance of vegetation (NIRv), a metric for canopy photosynthetic activity[28], was calculated from daily 3 m resolution normalized difference vegetation index (NDVI) and near-infrared radiation (NIR) was collected from Planet Labs satellite imagery[73–75]. Near-infrared reflectance of vegetation was also used as a proxy for plant inputs to soils given that up to 20% of C fixed by photosynthesis is released by root exudation[26,27].

Ryde is the major soil series found under alfalfa in the region, and is widespread across the Sacramento San Joaquin-Delta and along the central coast of California[76]. Ryde soils belong to the fine-loamy, mixed, superactive, thermic Cumulic Endoaquolls taxonomic class and are very deep, poorly drained soils formed in alluvium from mixed rock sources and decomposed vegetative matter[76]. Total soil C concentrations (mean ± standard error) were $5.26 \pm 0.02\%$ at 0–15 cm, $5.00 \pm 0.15\%$ at 15–30 cm, and $1.99 \pm 0.09\%$ at 30–60 cm[49]. Total soil $N$ concentrations were $0.38 \pm 0.003\%$ at 0–15 cm, $0.35 \pm 0.01\%$ at 15–30 cm, and $0.16 \pm 0.01\%$ at 30–60 cm[49].

## Automated chamber measurements

Surface fluxes of $N_2O$, $CH_4$, and $CO_2$ were measured continuously from January 2017 to February 2021 using an automated chamber system. The system consisted of nine opaque, automated gas flux chambers (eosAC, Eosense, Nova Scotia, Canada) connected to a multiplexer (eosMX, Eosense, Nova Scotia, Canada). The multiplexer allowed for dynamically signaled chamber deployment and routed gases to a cavity ring-down spectrometer (Picarro G2508, Santa Clara, CA, USA). Chambers were measured sequentially over a 10-min sampling period with a 1.5-min flushing period before and after each measurement.

Chambers were deployed in a $10 \times 10\,m$ grid design, with each chamber approximately 5 m from other chambers. Extended 15 cm soil collars were utilized to maintain measurement collection and ensure chambers were not inundated during irrigation or high rainfall events. Chambers were randomly assigned to either plant rows ($n = 5$) or inter-plant areas of bare soil ($n = 4$). Chambers remained in their original positions throughout the field campaign, except for short periods (<3 days) during field management activities (e.g., harvest, winter grazing). Foliage near chambers were minimally trimmed as needed between harvests if it inhibited chamber closure.

To determine chamber volume, chamber collar heights were measured approximately weekly and interpolated between measurements to account for changes in chamber height over time. Chamber volumes were also used to calculate the minimum detectable flux of 0.002 nmol $N_2O$ $m^{-2}$ $s^{-1}$, 0.06 nmol $CO_2$ $m^{-2}$ $s^{-1}$, and 0.002 nmol $CH_4$ $m^{-2}$ $s^{-1}$[77]. The minimum detectable fluxes reported here are conservative estimates, as the actual chamber volume was always smaller than the maximum theoretical volume used to calculate these values.

Flux calculations and analyses were first performed using Eosense eosAnalyze-AC v. 3.7.7 software, then data quality assessment and control were subsequently performed in R (RStudio, v.1.1.4633)[78]. Fluxes were removed from the final dataset if they were associated with erroneous spectrometer cavity temperature or pressure readings or if any gas concentrations were negative, corresponding to instrument malfunction. Fluxes were also removed if the chamber

deployment period was less than 9 min or greater than 11 min, indicative of chamber malfunction. Calculated linear and exponential fluxes were compared using estimate uncertainty to estimate ratios, and in cases where both the linear and exponential models produced high uncertainty, the individual flux was eliminated from the dataset. Data filtering removed 2.1% of flux measurement periods, generating a final dataset of 108,638, 103,013, and 102,997 simultaneous flux measurements of $CO_2$, $N_2O$, and $CH_4$, respectively. Following data filtering, all statistical analyses were performed using JMP Pro 15 (SAS Institute Inc., Cary, NC). Differences in site year, hourly, and seasonal mean flux values were analyzed with one-way ANOVAs followed by post-hoc Tukey tests. Values reported in the text are means ± standard errors unless otherwise noted.

To quantify site-level $CO_2$ uptake[49] and calculate site-level global warming potential (GWP) we utilized annual net ecosystem exchange (NEE) estimates from a nearby (<1 km) Ameriflux tower[72] in alfalfa grown with identical management practices and soil type. Here we used the eddy covariance technique[79] to capture continuous, long-term exchange of $CO_2$, $CH_4$, $H_2O$, and energy fluxes between the landscape and the atmosphere, along with measurements of environmental drivers[80]. Fluxes were measured at a frequency of 20 Hz using open-path infrared gas analyzers (LI-7500 for $CO_2$ and $H_2O$, LI-7700 for $CH_4$, LiCOR Inc., Lincoln, NE, USA) that were calibrated at least every 6 months. Sonic anemometers measured sonic temperature and three-dimensional wind speed at 20 Hz (WindMaster Pro 1590, Gill Instruments Ltd, Lymington, Hampshire, England). To convert $N_2O$ and $CH_4$ flux measurements to $CO_2e$, we used the IPCC AR5 100-year GWP values of 28 $CO_2e$ for $CH_4$ and 298 $CO_2e$ for $N_2O$[81].

## Quantifying hot moments of greenhouse gas emissions

This large, continuous dataset allowed us to quantify $N_2O$ hot moments and their impact on total $N_2O$ emissions. Following data filtering, the quantity and magnitude of hot moment measurements and their impact on annual flux estimates were determined. We defined hot moments as flux measurements with values greater than four standard deviations from the mean[12], as statistically 99.9% of the population should fall within four standard deviations of the mean. Yearly mean flux values were then calculated for only hot moment fluxes, the entire flux dataset, and the flux dataset with hot moment observations removed to determine the impact of outlier fluxes on annual greenhouse gas emissions. Given our large and continuous dataset, we could also compare mean fluxes with and without high flux events[12,82] to better quantify the importance of hot moments.

## Weekly soil measurements

Weekly soil samples (0–15 cm depth, $n = 10$ week$^{-1}$) were randomly collected with a 6 cm diameter soil auger within 30 m of the chamber array from April 2018 to May 2019. Soil samples were analyzed for gravimetric soil moisture, soil pH, and 2 M potassium chloride (KCl) extractable nitrate ($NO_3^-$) plus nitrite ($NO_2^-$) and ammonium ($NH_4^+$). For KCl extracts, we utilized a 5:1 ratio of 2 M KCl volume to oven dry equivalent (ODE) soil that were shaken for 1 h and subsequently filtered with Whatman Grade 1 filter paper[83]. The KCl extracts were then analyzed colorimetrically for $NH_4^+$ and $NO_3^-$ using an AQ300 analyzer (Seal Instruments, Mequon, WI). Soil moisture was determined gravimetrically by drying 10 g of field-fresh soil to a constant weight at 105 °C. Soil pH was measured in a slurry of 10 g of field-fresh soil in 10 mL of distilled deionized water[84].

## Soil sensor measurements

Two sets of soil sensors were installed from September 2018-February 2021 at depths of 10 cm, 30 cm, and 50 cm. This included SO-110 oxygen ($O_2$) and soil temperature sensors (Apogee Instruments,

Logan, UT) and CS616 moisture sensors (Campbell Scientific, Logan, UT) connected to a CR1000 datalogger (Campbell Scientific, Logan, UT) storing data at 15 min intervals. Sensors remained installed throughout the year. Erroneous data corresponding to sensor malfunction were removed from the dataset, which included 1.7% of soil moisture measurements and 3.4% of soil $O_2$ measurements. In total, there were 73 of 839 days missing during the soil sensor measurement period.

## Weekly soil depth gas samples

Two replicate soil gas samples were taken for $CO_2$, $CH_4$, and $N_2O$ at 10 cm, 30 cm, and 50 cm depths weekly from September 2018 to December 2019. Instrument grade stainless steel 1/8" tubing (Restek, Bellefonte, PA) was installed in parallel to the soil sensors above, with approximately 15 cm of tubing installed with multiple sampling holes parallel to the soil surface. Sampling septa (Restek, Bellefonte, PA) were installed in 1/8" Swagelok union (Swagelok, Solon, OH) permanently connected to the stainless-steel tubing. Septa were changed monthly. Gas samples were collected weekly with 30 ml BD syringes after first clearing the tubing dead volume. Short periods of soil inundation following extensive rainfall (March-April 2019) made it impossible to collect gas samples from some depths. Samples were stored in over-pressurized 20 ml glass vials with thick septa (Geomicrobial Technologies, Oechelata, OK) until manual sample injection analysis on a Shimadzu GC-34 (Shimadzu Corp., Tokyo, Japan). Generalized pairwise regression analyses were used to explore the relationships between measured soil atmosphere $CO_2$, $CH_4$, and $N_2O$ concentrations and surface soil $CO_2$, $CH_4$, and $N_2O$ fluxes.

## Wavelet coherence analysis

Wavelet coherence analysis was used to identify interactions between soil greenhouse gas fluxes, NIRv, and the soil variables ($O_2$, moisture, and temperature at 10, 30, and 50 cm) measured[12,85,86]. Wavelet coherence is a tool for comparing time series and is used to determine significance, causality and scale-emergent interactions between variables[16,80,87]. Wavelet coherence measures the cross-correlation between time series and allowed us to explore relationships between greenhouse gas fluxes and potential controls of NIRv, $O_2$, moisture, and temperature at daily, monthly, and annual timescales. Wavelet coherence is derived from two time series as a function of decomposed frequency (Wave.$xy$) and the wavelet power spectrum (Power.$x$, Power.$y$) of each individual time series:

$$\text{Coherence} = \frac{|\text{Wave}.xy|^2}{\text{Power}.x \cdot \text{Power}.y} \quad (1)$$

This approach allows for continuous wavelet-based analysis of univariate and bivariate time series, facilitating comparisons of time series data across scales, leads, and lags[86,88]. Missing data were replaced with zeroes to compute an unbiased estimator of the wavelet variance for time series with missing observations[86,89]. Statistical significance ($p$-value) was computed using 1000 Monte Carlo simulations. All wavelet decomposition and coherence calculations were conducted using the WaveletComp 1.1 R package[88].

## Reporting summary

Further information on research design is available in the Nature Portfolio Reporting Summary linked to this article.

## Data availability

The daily mean greenhouse gas fluxes (chamber $CO_2$, $CH_4$, $N_2O$, and eddy covariance $CO_2$), satellite-derived vegetation indices (NIRv), soil gas concentrations, and soil $O_2$, temperature, and moisture sensor data generated in this study and used to create the figures have been deposited the Dryad database: https://doi.org/10.6078/D1ZQ53.

## Code availability

The corresponding R code used in this study have been deposited in the Dryad database: https://doi.org/10.6078/D1ZQ53.

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

## Acknowledgements

We appreciate assistance, data availability, and feedback from Heather Dang, Tibisay Peréz, and numerous other members of the Silver and the Berkeley Biometeorology Labs at University of California, Berkeley. We thank Christine O'Connell for the initial code development for data filtering. This material is based upon work supported by California Department of Water Resources (Award 4600011240, T.L.A., W.L.S., D.D.B., J.G.V., D.J.S.), McIntire Stennis grant CA- B-ECO-7673-MS (W.L.S.), Breakthrough Strategies & Solutions (W.L.S.), V. Kann Rasmussen Foundation (W.L.S.), Oak Creek Foundation (W.L.S.), Jewish Community Foundation (W.L.S.), Northern Trust Foundation (W.L.S.), Trisons Foundation (W.L.S.), and the Delta Stewardship Council Delta Science Program under Grant No. 5298 and California Sea Grant College Program Project R/SF-89 (T.L.A.). The contents of this material do not necessarily reflect the views and policies of the Delta Stewardship Council or California Sea Grant, nor does mention of trade names or commercial products constitute endorsement or recommendation for use. We thank the California Department of Water Resources and the Metropolitan Water District of Southern California for research site access and soil sampling permission.

## Author contributions

T.L.A. wrote the original draft and created the visualizations, W.L.S. and D.D.B. conducted reviews and editing of the original draft, T.L.A. and W.L.S. conceptualized the paper, T.L.A., W.L.S., D.D.B., J.G.V., and D.J.S. contributed to the methodology and data investigation.

## Competing interests

The authors declare no competing interests.
