## [Peer Review File · Nature Communications]

REVIEWER COMMENTS

Reviewer #1 (Remarks to the Author):

Overview and general recommendations:

The submitted manuscript quantifies the contributions of soil nitrous oxide (N₂O) fluxes, and particularly from hot moments, to CO₂e balance for alfalfa agroecosystems. Using a novel dataset of multi-year, continuous, high-resolution chamber flux measurements coupled with belowground sensors and periodic soil sampling, the study highlights novel findings that hot moments of N₂O, although appearing in less than 2% of measurements, accounted for almost half of annual N₂O emissions and substantially offset the ecosystem carbon sink previously estimated for alfalfa. Consistent with recent literature, most hot moments occurred during soil re-wetting events, when anaerobic microsites may be generated, and were additionally modulated by temperature and by crop activities. Data collection, processing, and analytical methods are described well and are consistent with common practices of soil and ecosystem scientists.

My primary concern regarding the quality of the current manuscript lies in the contextualization of results to the broad readership of Nature Communications, and I believe this concern can be alleviated by addressing my comments below. I am not completely convinced of the relevance of the CH₄ measurements to the study and suggest a couple of different ways to integrate this dataset better with CO₂ and N₂O. I am also concerned that the broader impacts of this work to agroecosystems and/or to drylands are lost due to the focus on one particular crop species, albeit one that is grown widely. Finally, although the results are presented well, they don't fully match with the introductory material and hypothesis provided; I suggest expanding your hypothesis to include other drivers of N₂O and CO₂e that you measure and that are more holistic to addressing N₂O contributions to alfalfa C balance. I am confident that addressing these concerns will increase the quality of the manuscript and will accelerate its timeline to publication.

Specific comments:

Abstract: Since considerable text and one figure are dedicated to describing patterns and mechanisms of diel and seasonal fluxes of N₂O, I anticipated those to be reported somewhere in the abstract. I suggest a few words or short sentence to describe main findings.

Introduction: Given the title and focus of the paper, the connections between your trace gases of interest needs more explicit description. For example, CO₂ also produces respiration pulses during irrigation but at different timing and using different metabolic mechanisms, but ecosystem CO₂ fluxes are not described in the introduction. I suggest to more fully describe CO₂ fluxes to complement the description of CH₄ and round out the background of all three gases of interest, or to instead to describe CH₄ and CO₂ together as having hot moments of carbon or CO₂e exchange that may be enhanced by

N₂O flux and offset the non-hot-moment C sink. For a further biogeochemical perspective, there is considerable literature showing interactions between methanotrophy and N₂O production that could support the inclusion of CH₄ and N₂O data together, and why a soil CH₄ sink could lead to an N₂O source. This may be useful to return to in your discussion as well.

Results/Discussion: Since you described a hypothesis in your introduction, I would like to see a conclusion in the last paragraph (or elsewhere if more appropriate) about whether that hypothesis was supported. You address pieces of it throughout but I don't see them integrated explicitly. I also suggest moving your supplemental GWP results to the main text as I find those particularly compelling.

Ln 61: A word is missing in this sentence: "...patterns and associated [mechanisms maybe?] of CO₂..."

Ln 66-68: I was not expecting this hypothesis given the diversity of measurements reported here; particularly, I would have expected NH₄⁺ and O₂ to be included as hypothesized drivers of N₂O emission since they were described earlier in the introduction. Is there a reason these were not included but NO₃⁻ was? Temperature/climatic drivers of emissions are not really introduced in the introduction and do not appear in hypotheses but are contained in multiple figures and results; consider including them here and in the intro at large.

Ln 103: A word is missing in this sentence: "...from greater [?] and C and substrate availability..."

Ln 184: I suggest including this as a new subsection titled "Aboveground C dynamics" or "Alfalfa C sequestration" as this paragraph includes alfalfa C sink and eddy covariance data rather than just soil CO₂ emissions. Or, make the heading of Ln 173 more inclusive as "Agroecosystem CO₂ balance" or similar to capture both above and belowground fluxes.

Ln 200: I suggest including this as a new subsection titled "synthesis" or "conclusions" as it integrates all data rather than just CO₂ fluxes.

Ln 216: How much land area is used for alfalfa? This information, with included citation, would be useful for upscaling.

Ln 341: Suggest to replace "gappy" with "discontinuous" or "noncontinuous."

Ln 549: Repository information needed (e.g. website link or citation).

Figures 1, 2, and 4 contain data that is too inter-connected and sometimes redundant to be stand-alone figures. It is not immediately clear to me why they are separate, so I suggest combining into one seven-panel figure. Alternatively, I would suggest pulling soil NH_4^+ and NO_3^- data from Fig 4 and 2, respectively, into their own figure, since these were not sensor-based measurements and took place over a shorter time period, and combine all sensor-based measurements.

Figure 5: More specific y-axis labels and caption description needed. For (A), is CO_2 flux from soil or whole ecosystem? I'm assuming these are just respiration fluxes and not NEE but please correct if I am wrong.

Figure 5: I would be curious, either as an additional panel in Figure 5 or a supplemental figure, what CO_2e flux (either $\text{CO}_2+\text{CH}_4+\text{N}_2\text{O}$ or just CO_2+CH_4 for carbon exchange) patterns look over time and how those relate to temp and NIRv.

Table 2: Do the + signs in hot moment % of total flux add additional information? To me, a percentage alone seems sufficient.

Figures S2, S6, S7: Since these figures are relaying similar types of information, suggest to standardize axis label font size and other figure labels. Similarly, standardize wavelet figures.

Reviewer #2 (Remarks to the Author):

Review of 'Hot moments of nitrous oxide emissions lower the carbon-sink potential of alfalfa Agriculture' by Anthony et al.

The potential offsetting of soil carbon sequestration through N_2O emissions from an irrigated alfalfa crop is the focus of this study. Eddy covariance measurements were conducted over 4 years capturing several emission events (hot moments) associated with irrigation. The methodology is sound, results are clearly presented for the most part and the manuscript is well written. However, I have two main concerns:

- 1) the interpretation of the N₂O flux results and its generalization to 'alfalfa agriculture';
- 2) a lack of clear explanation of what the 'novel sensor suite' is exactly, and how it provided integrated information that improved our understanding of GHG fluxes.

I am providing detailed comments on each of these concerns below.

- 1) the interpretation of the N₂O flux results and its generalization to 'alfalfa agriculture';

The hypothesis is "that elevated NO₃ concentrations and irrigation during the growing season would stimulate hot moments of N₂O emission, offsetting a significant portion of the net CO₂-equivalent (CO₂e) sink." Alfalfa is a perennial legume crop and hence nitrate levels would be expected to be minimized during the growing season in comparison to annual crops receiving N inputs in the form of manure or synthetic fertilizer such as corn or wheat. In fact, increased use of perennials is recognized as an N₂O-mitigating practice.

I was at first surprised by the hypothesis statement, but results of high N₂O emissions were corroborated with measurements, surely indicating elevated NO₃ levels likely from soil organic matter mineralization. A closer look at the site's soil characteristics such as C levels indicates this may be an organic soil (or at least has very high levels of C), which are well known to have very high N₂O emissions. In fact, Pärn et al. (2018) identify N rich organic soils under well-drained conditions as global N₂O hot spots.

Pärn et al. (2018) refer to a US-CA hot-spot based on a sampling site at Lat. 38.0169 N, Long. -121.6203 W, close to the experimental site of 38.11°N, 121.5°W. Anthony and Silver (2021) did measure N₂O fluxes with automated chambers at what appears to be an adjacent site to this study also identifying this area as a hot-spot of N₂O fluxes (in this case the site had soil C values about 3x as reported here). However, in this manuscript the relatively high C levels of the site are not brought up as a potential major driver.

In addition, it appears that the Ryde soil series does have relatively low pH (not reported in manuscript). However, the role of soil organic matter and of the low pH in driving high soil N₂O production is not considered in the interpretation of N₂O flux results. Rather, the focus is on the role of alfalfa plants.

I think it is very difficult to separate the two effects (soil vs. plants) based on the set of measurements conducted, even though the authors collected a relatively long and complete time series of N₂O fluxes. The main short-coming is the lack of a comparison (fallow field vs. alfalfa?, annual unfertilized crop vs.

alfalfa?) that would allow for teasing out the main drivers. I expect a fallow or annual crop, irrigated field at this site would see even higher N₂O fluxes than observed due to the high carbon and nitrogen substrates fueling denitrification and the acidic conditions favoring a higher N₂O/N₂ ratio. This means that alfalfa could actually be working to decrease emissions under these very favorable conditions for N₂O production. Although measurements in Anthony and Silver (2020) were made at a different sites (and different soils) in the same area, it would be interesting to compare these datasets.

I also miss an interpretation of the unique conditions of the experimental site vis-à-vis areas where alfalfa is grown in the US or the world. For example, the statement “Our results show that N₂O emissions can significantly lower the C sink potential of this globally important crop” (L. 206) implies that experimental results could perhaps be applicable to other areas and any caveats are not identified. Instead, it is suggested that the high N₂O fluxes are due to the intensive sampling (see L. 72 “Annual mean N₂O fluxes were 624.4 ± 26.8 mg N₂O m⁻² yr⁻¹ (Table 1, range: 247.0 ± 5.7 to 901.9 ± 74.5 mg N₂O m⁻² yr⁻¹) and were significantly greater than other N₂O flux estimates in alfalfa using less intensive periodic sampling (16–19)”) and the unique soil conditions are not considered.

2) The lack of clear explanation of what the ‘novel sensor suite’ is exactly, and how it provided integrated information that improved our understanding of GHG fluxes.

Several times reference is made to a ‘novel sensor suite’. For example, in the Abstract: “Using a novel suite of continuous soil sensing, eddy covariance, and satellite imagery we found that N₂O emissions offset the ecosystem greenhouse gas sink by up to 14% annually” and L. 23 “Data from this novel sensor suite show that N₂O emissions significantly lower the carbon-sink potential of alfalfa agriculture.”

However, the main finding of a 14% emission offset is based on the eddy covariance measurements alone. Although, EC measurements of N₂O fluxes are not nearly as widespread as for CO₂ and H₂O, or even CH₄, I would not classify the EC technique as novel. Similar high temporal resolution data as presented here can also be derived with automated chambers as Anthony and Silver (2020) have done for an adjacent site, but do not have the advantage of fluxes being spatially integrated over large areas like EC has.

The continuous soil sensing was used to provide some explanation for the flux drivers as is often done in N₂O studies. The satellite imagery was used to find “Significant coherence between satellite derived vegetation growth and N₂O fluxes suggested that plant activity was an important driver of background emissions” (Abstract) and “Background fluxes varied with moisture, temperature, and NIRv, an index of GPP. Lagged relationships between NIRv, CO₂, and N₂O fluxes suggested that plant inputs were likely an important driver of soil CO₂ fluxes and background N₂O emissions.” (L. 204). I consider the latter conclusion a bit tenuous since plant inputs (presumably substrate for nitrifiers and denitrifiers) were not specifically quantified and compared to soil substrate supply. In addition, this relationship was only

found for background N₂O emissions while as described in this manuscript, hot moments are the main overall drivers of N₂O emissions.

References

T. L. Anthony, W. L. Silver, Mineralogical associations with soil carbon in managed wetland soils. *Glob. Chang. Biol.* 26, 6555–6567 (2020).

Pärn et al. Nitrogen-rich organic soils under warm well-drained conditions are global nitrous oxide emission hotspots. *Nature Communications* (2018) 9:1135, DOI 10.1038/s41467-018-03540-1,

RESPONSE TO REFEREES

Reviewer #1 (Remarks to the Author):

Overview and general recommendations:

The submitted manuscript quantifies the contributions of soil nitrous oxide (N₂O) fluxes, and particularly from hot moments, to CO₂e balance for alfalfa agroecosystems. Using a novel dataset of multi-year, continuous, high-resolution chamber flux measurements coupled with belowground sensors and periodic soil sampling, the study highlights novel findings that hot moments of N₂O, although appearing in less than 2% of measurements, accounted for almost half of annual N₂O emissions and substantially offset the ecosystem carbon sink previously estimated for alfalfa. Consistent with recent literature, most hot moments occurred during soil re-wetting events, when anaerobic microsites may be generated, and were additionally modulated by temperature and by crop activities. Data collection, processing, and analytical methods are described well and are consistent with common practices of soil and ecosystem scientists.

My primary concern regarding the quality of the current manuscript lies in the contextualization of results to the broad readership of Nature Communications, and I believe this concern can be alleviated by addressing my comments below. I am not completely convinced of the relevance of the CH₄ measurements to the study and suggest a couple of different ways to integrate this dataset better with CO₂ and N₂O. I am also concerned that the broader impacts of this work to agroecosystems and/or to drylands are lost due to the focus on one particular crop species, albeit one that is grown widely. Finally, although the results are presented well, they don't fully match with the introductory material and hypothesis provided; I suggest expanding your hypothesis to include other drivers of N₂O and CO₂e that you measure and that are more holistic to addressing N₂O contributions to alfalfa C balance. I am confident that addressing these concerns will increase the quality of the manuscript and will accelerate its timeline to publication.

We thank the reviewer for their comments. We appreciate the suggestions to broaden the contextualization and better integrate the CO₂ and CH₄ data and have made the detailed changes suggested (outlined below). We have also expanded the first hypothesis as suggested and added a second hypothesis that states, "We also hypothesized that background patterns in N₂O emissions would follow patterns in plant activity indicative of potential changes in C or substrate availability." Line numbers cited below represent numbers with track changes on.

Specific comments:

Abstract: Since considerable text and one figure are dedicated to describing patterns and mechanisms of diel and seasonal fluxes of N₂O, I anticipated those to be reported somewhere in the abstract. I suggest a few words or short sentence to describe main findings.

We have added text regarding diel and seasonal N₂O fluxes (L22-L23).

Introduction: Given the title and focus of the paper, the connections between your trace gases

of interest needs more explicit description. For example, CO₂ also produces respiration pulses during irrigation but at different timing and using different metabolic mechanisms, but ecosystem CO₂ fluxes are not described in the introduction. I suggest to more fully describe CO₂ fluxes to complement the description of CH₄ and round out the background of all three gases of interest, or to instead to describe CH₄ and CO₂ together as having hot moments of carbon or CO₂e exchange that may be enhanced by N₂O flux and offset the non-hot-moment C sink. For a further biogeochemical perspective, there is considerable literature showing interactions between methanotrophy and N₂O production that could support the inclusion of CH₄ and N₂O data together, and why a soil CH₄ sink could lead to an N₂O source. This may be useful to return to in your discussion as well.

Additional text regarding CO₂ and CH₄ fluxes was added in the introduction (L61-L77) and discussion (L213-216). We also now mention that eddy covariance studies suggest that alfalfa agroecosystems are net C sinks even with potential pulses of CO₂ and CH₄ emissions, but data on continuous N₂O emissions are lacking.

Results/Discussion: Since you described a hypothesis in your introduction, I would like to see a conclusion in the last paragraph (or elsewhere if more appropriate) about whether that hypothesis was supported. You address pieces of it throughout but I don't see them integrated explicitly. I also suggest moving your supplemental GWP results to the main text as I find those particularly compelling.

We have added a conclusions section and made the suggested changes which we agree really help highlight the findings and significance. (L248-254). We also integrated the supplemental GWP results to the main text as suggested (L110-117).

Ln 61: A word is missing in this sentence: "...patterns and associated [mechanisms maybe?] of CO₂..."

We have added the word "controls".

Ln 66-68: I was not expecting this hypothesis given the diversity of measurements reported here; particularly, I would have expected NH₄⁺ and O₂ to be included as hypothesized drivers of N₂O emission since they were described earlier in the introduction. Is there a reason these were not included but NO₃⁻ was? Temperature/climatic drivers of emissions are not really introduced in the introduction and do not appear in hypotheses but are contained in multiple figures and results; consider including them here and in the intro at large.

We expanded the hypotheses as suggested and added some text to the introduction to better justify the hypotheses and provide background for the measurements made to test the hypotheses.

Ln 103: A word is missing in this sentence: "...from greater [?] and C and substrate availability..."

This has been changed to “greater C and N substrate availability” (L140)

Ln 184: I suggest including this as a new subsection titled “Aboveground C dynamics” or “Alfalfa C sequestration” as this paragraph includes alfalfa C sink and eddy covariance data rather than just soil CO₂ emissions. Or, make the heading of Ln 173 more inclusive as “Agroecosystem CO₂ balance” or similar to capture both above and belowground fluxes.

We have changed the heading to “Agroecosystem CO₂ balance” (L218)

Ln 200: I suggest including this as a new subsection titled “synthesis” or “conclusions” as it integrates all data rather than just CO₂ fluxes.

We added a conclusions section as suggested.

Ln 216: How much land area is used for alfalfa? This information, with included citation, would be useful for upscaling.

We clarify that alfalfa and corn are the dominant agricultural land uses in the region, with alfalfa representing 20% of agricultural land area in the Sacramento-San Joaquin Delta (The Delta Protection Commission, 2020) and the largest crop by area in California and the (Putnam et al. 2007). Nearly 100% of alfalfa in California is irrigated (Putnam et al. 2007). (L270-L274).

Ln 341: Suggest to replace “gappy” with “discontinuous” or “noncontinuous.”

We have replaced “gappy time series” with “time series with missing observations” (L400).

Ln 549: Repository information needed (e.g. website link or citation).

Repository information has been added as Source Data and as:

<https://datadryad.org/stash/share/igfrCACBTOMTNEi8KsL3auVLnSqKiN51WFRUIF04Ds>.

Figures 1, 2, and 4 contain data that is too inter-connected and sometimes redundant to be stand-alone figures. It is not immediately clear to me why they are separate, so I suggest combining into one seven-panel figure. Alternatively, I would suggest pulling soil NH₄⁺ and NO₃⁻ data from Fig 4 and 2, respectively, into their own figure, since these were not sensor-based measurements and took place over a shorter time period, and combine all sensor-based measurements.

Figure 5: More specific y-axis labels and caption description needed. For (A), is CO₂ flux from soil or whole ecosystem? I’m assuming these are just respiration fluxes and not NEE but please correct if I am wrong.

Figure 5: I would be curious, either as an additional panel in Figure 5 or a supplemental figure, what CO₂e flux (either CO₂+CH₄+N₂O or just CO₂+CH₄ for carbon exchange) patterns look over

time and how those relate to temp and NIRv.

We have addressed the suggestions above and combined and rearranged the figures: Figure 1 contains N₂O, CH₄, and CO₂ fluxes with soil moisture, temperature, O₂, and NIRv. Figure 3 now contains NO₃, NH₄, and soil pH observations.

Table 2: Do the + signs in hot moment % of total flux add additional information? To me, a percentage alone seems sufficient.

We have removed the + signs from Table 2.

Figures S2, S6, S7: Since these figures are relaying similar types of information, suggest to standardize axis label font size and other figure labels. Similarly, standardize wavelet figures.

We have standardized font sizes across the supplemental figures (soil GHG concentrations and wavelet figures).

Reviewer #2 (Remarks to the Author):

Review of 'Hot moments of nitrous oxide emissions lower the carbon-sink potential of alfalfa Agriculture' by Anthony et al.

The potential offsetting of soil carbon sequestration through N₂O emissions from an irrigated alfalfa crop is the focus of this study. Eddy covariance measurements were conducted over 4 years capturing several emission events (hot moments) associated with irrigation. The methodology is sound, results are clearly presented for the most part and the manuscript is well written. However, I have two main concerns:

- 1) the interpretation of the N₂O flux results and its generalization to 'alfalfa agriculture';
- 2) a lack of clear explanation of what the 'novel sensor suite' is exactly, and how it provided integrated information that improved our understanding of GHG fluxes.

I am providing detailed comments on each of these concerns below.

We thank the reviewer for their comments and address them specifically below.

- 1) the interpretation of the N₂O flux results and its generalization to 'alfalfa agriculture';

The hypothesis is "that elevated NO₃ concentrations and irrigation during the growing season would stimulate hot moments of N₂O emission, offsetting a significant portion of the net CO₂-equivalent (CO₂e) sink." Alfalfa is a perennial legume crop and hence nitrate levels would be expected to be minimized during the growing season in comparison to annual crops receiving N inputs in the form of manure or synthetic fertilizer such as corn or wheat. In fact, increased use of perennials is recognized as an N₂O-mitigating practice.

We agree that the assumption is often that an N-fixing perennial crop does not experience hot moments of NO_3^- availability or N_2O emissions. However, the high-resolution measurements we were able to achieve showed that NO_3^- concentrations were not uniformly low, and this together with periods of high soil moisture led to hot moments of N_2O emissions. We have added text in the introduction to better summarize these issues (L88-95) and expanded the hypotheses and explanations as suggested by reviewer one.

I was at first surprised by the hypothesis statement, but results of high N_2O emissions were corroborated with measurements, surely indicating elevated NO_3^- levels likely from soil organic matter mineralization. A closer look at the site's soil characteristics such as C levels indicates this may be an organic soil (or at least has very high levels of C), which are well known to have very high N_2O emissions. In fact, Pärn et al. (2018) identify N rich organic soils under well-drained conditions as global N_2O hot spots.

These are degraded peatland soils, and thus would not typically be classified as an organic soil (e.g., > 12% C frequently as suggested in papers such as Pärn et al. (2018). However, we agree that soil organic matter mineralization could play a role in addition to N added via N-fixation. For example, mineralized alfalfa roots and shoots can be a significant source NO_3^- (up to 75 kg N- $\text{NO}_3^- \text{ ha}^{-1}$, Kavdir et al. 2005). We have added text to this effect (L269-274).

Pärn et al. (2018) refer to a US-CA hot-spot based on a sampling site at Lat. 38.0169 N, Long. -121.6203 W, close to the experimental site of 38.11°N, 121.5°W. Anthony and Silver (2021) did measure N_2O fluxes with automated chambers at what appears to be an adjacent site to this study also identifying this area as a hot-spot of N_2O fluxes (in this case the site had soil C values about 3x as reported here). However, in this manuscript the relatively high C levels of the site are not brought up as a potential major driver.

We now mention soil organic matter can be a potential driver of greenhouse gas emissions but note that this site has lower soil C and N than others in the regions (L145-146, L220). The three sites in the Sacramento-San Joaquin Delta described above have widely ranging surface soil C concentrations (5-20%) and inorganic N concentrations (1-300 $\mu\text{g N g soil}^{-1}$), are on different soil series (Parn et al. 2018: Shima, Anthony and Silver 2021: Rindge, and this study: Ryde) and occur under differing land management practices (pasture, corn, alfalfa).

In addition, it appears that the Ryde soil series does have relatively low pH (not reported in manuscript). However, the role of soil organic matter and of the low pH in driving high soil N_2O production is not considered in the interpretation of N_2O flux results. Rather, the focus is on the role of alfalfa plants.

We agree the low pH can be a contributor to the magnitude of hot moments. Soil pH was measured weekly alongside soil mineral N values (added as Figure 2c) and relevant text has been added clarifying the effects of an acidic soil pH (L53-55, L143-146).

I think it is very difficult to separate the two effects (soil vs. plants) based on the set of

measurements conducted, even though the authors collected a relatively long and complete time series of N₂O fluxes. The main short-coming is the lack of a comparison (fallow field vs. alfalfa?, annual unfertilized crop vs. alfalfa?) that would allow for teasing out the main drivers. I expect a fallow or annual crop, irrigated field at this site would see even higher N₂O fluxes than observed due to the high carbon and nitrogen substrates fueling denitrification and the acidic conditions favoring a higher N₂O/N₂ ratio. This means that alfalfa could actually be working to decrease emissions under these very favorable conditions for N₂O production. Although measurements in Anthony and Silver (2020) were made at a different sites (and different soils) in the same area, it would be interesting to compare these datasets.

We understand the reviewer's comments, but respectfully disagree that a comparison to other crops or a fallowed field is needed to determine the main drivers of N₂O in this context. Here we used wavelet coherence to parse relationships among variables measured. It is a useful tool for studying fluctuations across data-rich time series and can be used to determine significance and elucidate scale-emergent interactions between variables (Chamberlain et al. 2018, Rodríguez-Murillo and Filella, 2020, Sturtevant et al., 2016). A more detailed description of wavelet coherence has been added to the text (L389-L392). Comparisons with a fallowed field or irrigated annual crop, while interesting, would be asking a different question than the one addressed here. Fallow and annual cropping are different management activities that change multiple environmental and biogeochemical conditions when compared to alfalfa agriculture. Annual cropping is conducted on different soils. Thus, we don't feel that a comparison with corn cropping would help us determine the impacts of alfalfa on N₂O emissions. In this instance we explored the potential role of plant productivity in greenhouse gas fluxes when compared with other drivers occurring under the same environmental conditions. We now clarify and qualify this result more carefully to address the reviewer's concerns. We included text stating that these degraded peatland soils that have lost a significant proportion of their initial organic matter and contain significantly higher mineral content than intact or nutrient-rich peatland soils by citing Anthony and Silver (2020). We have added text throughout to clarify these issues.

I also miss an interpretation of the unique conditions of the experimental site vis-à-vis areas where alfalfa is grown in the US or the world. For example, the statement "Our results show that N₂O emissions can significantly lower the C sink potential of this globally important crop" (L. 206) implies that experimental results could perhaps be applicable to other areas and any caveats are not identified. Instead, it is suggested that the high N₂O fluxes are due to the intensive sampling (see L. 72 "Annual mean N₂O fluxes were 624.4 ± 26.8 mg N₂O m⁻² yr⁻¹ (Table 1, range: 247.0 ± 5.7 to 901.9 ± 74.5 mg N₂O m⁻² yr⁻¹) and were significantly greater than other N₂O flux estimates in alfalfa using less intensive periodic sampling (16–19)") and the unique soil conditions are not considered.

We agree that more information regarding our site in relation to alfalfa agriculture is warranted and has been included (L31-L33, L50-L51, L267-L274). These are highly degraded peatland soils that have lost a significant proportion of their initial organic matter content and contains significantly higher mineral concentrations than intact or nutrient-rich peatland

soils (Anthony and Silver 2020). We clarify that the management of the site is representative of alfalfa agriculture across the region as nearly 100% of alfalfa in California is irrigated, and approximately 82% is similarly flood irrigated (Long et al. 2022). We have added text (L272-L273) comparing our site to general irrigation management and soil types, adding caveats regarding soil pH (L143-L145, L170-L172).

In L121-L125 we also clarify why we believe intensive sampling is important. Not missing these hot moments (which represent <1% of observations but an average of 44% of emissions) more accurately captures short-term hot moment N₂O emissions where non-continuous sampling methods can miss rare high flux events.

2) The lack of clear explanation of what the ‘novel sensor suite’ is exactly, and how it provided integrated information that improved our understanding of GHG fluxes.

Several times reference is made to a ‘novel sensor suite’. For example, in the Abstract: “Using a novel suite of continuous soil sensing, eddy covariance, and satellite imagery we found that N₂O emissions offset the ecosystem greenhouse gas sink by up to 14% annually” and L. 23 “Data from this novel sensor suite show that N₂O emissions significantly lower the carbon-sink potential of alfalfa agriculture.”

However, the main finding of a 14% emission offset is based on the eddy covariance measurements alone. Although, EC measurements of N₂O fluxes are not nearly as widespread as for CO₂ and H₂O, or even CH₄, I would not classify the EC technique as novel. Similar high temporal resolution data as presented here can also be derived with automated chambers as Anthony and Silver (2020) have done for an adjacent site, but do not have the advantage of fluxes being spatially integrated over large areas like EC has.

We agree clarification in the novel sensor suite is needed and believe this caused some of the misunderstanding described here. We have added text clarifying that this study used “a novel suite of automated flux chambers (to quantify continuous CO₂, N₂O and CH₄ fluxes) combined with continuous soil sensing, eddy covariance (to quantify net ecosystem CO₂ exchange) , and satellite imagery” in the abstract (L18-19) and introduction (L81-L83). We do not suggest that EC, even with N₂O, is novel, rather the novelty is provided by the combination of long-term automated chamber, continuous soil sensing, satellite imagery, and eddy covariance observations.

The continuous soil sensing was used to provide some explanation for the flux drivers as is often done in N₂O studies. The satellite imagery was used to find “Significant coherence between satellite derived vegetation growth and N₂O fluxes suggested that plant activity was an important driver of background emissions” (Abstract) and “Background fluxes varied with moisture, temperature, and NIRv, an index of GPP. Lagged relationships between NIRv, CO₂, and N₂O fluxes suggested that plant inputs were likely an important driver of soil CO₂ fluxes and background N₂O emissions.” (L. 204). I consider the latter conclusion a bit tenuous since plant inputs (presumably substrate for nitrifiers and denitrifiers) were not specifically quantified

and compared to soil substrate supply. In addition, this relationship was only found for background N₂O emissions while as described in this manuscript, hot moments are the main overall drivers of N₂O emissions.

Here we used wavelet coherence to compare variables across time series and infer relationships among variables. We have added text in the introduction regarding substrate availability (L65-71) and clarified that our plant activity metric (NIRv) specifically represents photosynthetic activity (L24-25, L82-L83, L277-282). NIRv is an accurate method to determine canopy photosynthesis (Baldocchi et al. 2020), and photosynthesis is the primary source of C inputs into terrestrial ecosystems. Root exudates are well-known labile soil C sources that can prime microbial activity (Panchal et al. 2022), with up to 20% of C fixed by photosynthesis released by root exudation (Guyonnet et al. 2018, Haichar et al., 2008). Thus, the relationships with plant photosynthetic activity are an index of plant C inputs and activity (L65-L70, L280-L282). We have clarified that the observed lagged relationships may represent delays between photosynthetic C uptake and root exudation processes (L242-244). We have also softened the language to suggest that these patterns are one possible driver of the background patterns observed.

Hot moments of emissions appear to have been driven by O₂, and moisture at daily scales, and that lagged (weekly to monthly) relationships between NIRv and N₂O fluxes suggests plant inputs were likely an important driver background N₂O emissions (L176-L187).

References

- T. L. Anthony, W. L. Silver, Mineralogical associations with soil carbon in managed wetland soils. *Glob. Chang. Biol.* 26, 6555–6567 (2020).
- Pärn et al. Nitrogen-rich organic soils under warm well-drained conditions are global nitrous oxide emission hotspots. *Nature Communications* (2018) 9:1135, DOI 10.1038/s41467-018-03540-1,

Response to reviewer references:

T. L. Anthony, W. L. Silver, Mineralogical associations with soil carbon in managed wetland soils. *Glob. Chang. Biol.* 26, 6555–6567 (2020).

D. D. Baldocchi, Y. Ryu, B. Dechant, E. Eichelmann, K. Hemes, S. Ma, C. R. Sanchez, R. Shortt, D. Szutu, A. Valach, J. Verfaillie, G. Badgley, Y. Zeng, J. A. Berry, Outgoing Near-Infrared Radiation From Vegetation Scales With Canopy Photosynthesis Across a Spectrum of Function, Structure, Physiological Capacity, and Weather. *J. Geophys. Res. Biogeosciences.* 125 (2020), doi:10.1029/2019JG005534.

Chamberlain, Samuel D., Tyler L. Anthony, Whendee L. Silver, Elke Eichelmann, Kyle S. Hemes, Patricia Y. Oikawa, Cove Sturtevant, Daphne J. Szutu, Joseph G. Verfaillie, and Dennis D. Baldocchi. "Soil Properties and Sediment Accretion Modulate Methane Fluxes from Restored

Wetlands.” *Global Change Biology* 24, no. 9 (2018): 4107–21.
<https://doi.org/10.1111/gcb.14124>.

The Delta Protection Commission. “The State of Delta Agriculture : Economic Impact , Conservation and Trends,” 2020.

Guyonnet, Julien P., Amélie A. M. Cantarel, Laurent Simon, and Feth el Zahar Haichar. “Root Exudation Rate as Functional Trait Involved in Plant Nutrient-Use Strategy Classification.” *Ecology and Evolution* 8, no. 16 (2018): 8573–81. <https://doi.org/10.1002/ece3.4383>.

Haichar, Feth el Zahar, Christine Marol, Odile Berge, J. Ignacio Rangel-Castro, James I. Prosser, Jérôme Balesdent, Thierry Heulin, and Wafa Achouak. “Plant Host Habitat and Root Exudates Shape Soil Bacterial Community Structure.” *The ISME Journal* 2, no. 12 (2008): 1221–30.
<https://doi.org/10.1038/ismej.2008.80>.

Kavdir, Yasemin, Daniel P. Rasse, and Alvin J. M. Smucker. “Specific Contributions of Decaying Alfalfa Roots to Nitrate Leaching in a Kalamazoo Loam Soil.” *Agriculture, Ecosystems & Environment* 109, no. 1 (2005): 97–106. <https://doi.org/10.1016/j.agee.2005.02.020>.

Long RF, Goodell PB, Baldwin RA, Frate CA, Godfrey LD, Orloff SB, Canevari WM, Davis RM, Natwick ET, Putnam DH, Westerdahl BB. Revised continuously. UC IPM Pest Management Guidelines: Alfalfa. UC ANR Publication 3430. Oakland, CA.

Panchal, Poonam, Catherine Preece, Josep Peñuelas, and Jitender Giri. “Soil Carbon Sequestration by Root Exudates.” *Trends in Plant Science* 27, no. 8 (2022): 749–57.
<https://doi.org/10.1016/j.tplants.2022.04.009>.

Putnam, D. H.; Summers, C. G.; Orloff, S. B. “Alfalfa Production Systems in California.” In *Irrigated Alfalfa Management for Mediterranean and Desert Zones*. Oakland, California: University of California Agriculture and Natural Resources, 2007.
<http://alfalfa.ucdavis.edu/IrrigatedAlfalfa>.

Rodríguez-Murillo, Juan Carlos, and Montserrat Filella. “Significance and Causality in Continuous Wavelet and Wavelet Coherence Spectra Applied to Hydrological Time Series.” *Hydrology* 7, no. 4 (2020): 82. <https://doi.org/10.3390/hydrology7040082>.

Sturtevant, Cove, Benjamin Ruddell, Sara Helen Knox, Joseph Verfaillie, Jaclyn Hatala Matthes, Patricia Y Oikawa, and Dennis Baldocchi. “Identifying Scale-Emergent, Nonlinear, Asynchronous Processes of Wetland Methane Exchange.” *Journal of Geophysical Research: Biogeosciences*, 2016, 188–204. <https://doi.org/10.1002/2015JG003054.Received>.

REVIEWER COMMENTS

Reviewer #1 (Remarks to the Author):

Overview and general recommendations:

The submitted manuscript quantifies the contributions of soil nitrous oxide (N₂O) fluxes, and particularly from hot moments, to CO₂e balance for alfalfa agroecosystems. Using a novel dataset of multi-year, continuous, high-resolution chamber flux measurements coupled with belowground sensors and periodic soil sampling, the study highlights novel findings that hot moments of N₂O, although appearing in less than 2% of measurements, accounted for almost half of annual N₂O emissions and substantially offset the ecosystem carbon sink previously estimated for alfalfa. Consistent with recent literature, most hot moments occurred during soil re-wetting events, when anaerobic microsites may be generated, and were additionally modulated by temperature and by crop activities. Data collection, processing, and analytical methods are described well and are consistent with common practices of soil and ecosystem scientists.

After reading responses to reviewers and the revised manuscript, I am pleased with the changes the authors have made and I think revisions have made clear the importance and nuance of these results. I have a few comments regarding minor detail changes, but I think generally this manuscript is ready to move forward to publication. Great work!

Specific comments:

Abstract: I think somewhere in here you need to mention CO₂ and CH₄ as measured carbon fluxes since those are a big chunk of your results and you spend a lot of text describing them. The least invasive way to include them might be in Ln 18-19: "...offsetting the ecosystem carbon (CO₂ and CH₄) sink by ...". Or something more elegant. I think it's important to specify somehow that you looked at CH₄ in addition to CO₂, since readers may assume only CO₂ was measured here.

Ln 38-39: Citations needed.

Ln 59-60: This citation does not have a matching reference in References section and is different annotation than other citations.

Ln 163-165: I'm getting confused by the wording here; can you write this sentence more simply? Is it that lower GPP tended to match low background N₂O?

Reviewer #2 (Remarks to the Author):

Thank you for addressing many of my comments and clarifying certain issues. I still have major concerns as to how the results are interpreted and conveyed to readers. The main take away message is that 'alfalfa agriculture' emits significant amounts of N₂O, enough to offset the C-sink potential. The high overall annual N₂O emissions (=4 kg N/ha/yr with range of 1.6 to 5.7) occur over somewhat frequent and short periods of time associated with irrigation and driven by high N levels released by the high SOM content at the measurement site (Hot moments). I commented on the high SOM in my previous review and although the reply was that the site is on highly degraded peatland soil, the SOM content is still quite high compared to regular mineral soils. This means that the results obtained here can not be easily generalized to 'alfalfa agriculture' as the authors have done in Abstract and Conclusions. A much more nuanced interpretation of the measurements is needed. In fact, N₂O emissions are notoriously variable with soil type, management and weather events and generalizing results from one site is not possible. For example, Tenuta et al. (2019) using a micromet approach show that including perennial crops such as alfalfa in crop rotations leads to much lower N₂O emissions than annual crops, in contrast to this study.

In addition, the authors argue the large N₂O emissions measured in this study are due to the measurement method used. For example in L. 89-91 "Annual mean N₂O fluxes were 624.4 ± 26.8 mg and were significantly greater than other N₂O flux estimates in alfalfa using less intensive periodic sampling^{30–33}". While I strongly agree that more continuous measurements as provided by micromet methods are sorely needed and provide a much better flux time series, I think a more nuanced interpretation is also needed here. Firstly, the citations given here are not appropriate to support the argument made.

Ref 30: reports on emissions that are of a similar order of magnitude (2.3 and 5.7 kg N/ha/yr) with measurements made using static chambers;

Ref 32: is a micromet study with frequent observations;

Ref 33: reports on canola and wheat N₂O emissions following alfalfa termination for a semi-arid region.

Secondly, as demonstrated by Ref30 the high emission peaks could be captured with other methods in an irrigated alfalfa system since they are quite predictable and timed with the irrigation. In fact, there may be an issue with overestimation given that chamber studies will typically target high emission events and then interpolate between these weekly data points to obtain annual emissions.

The final point of concern (which I missed in my first review) is the lack of consideration in C removal in harvest when determining if the field is a carbon sink. In fact, some information about forage yields and C content should be included.

References

Tenuta, M., Amiro, B. D., Gao, X. P., Wagner-Riddle, C., Gervais, M. (2019). Agricultural management practices and environmental drivers of nitrous oxide emissions over a decade for an annual and an annual-perennial crop rotation. *Agricultural and Forest Meteorology*, 276, Article 107636. <https://doi.org/10.1016/j.agrformet.2019.107636>

REVIEWER COMMENTS

Responses by the authors are in bold. Corresponding line numbers refer to line numbers with track changes on.

Reviewer #1 (Remarks to the Author):

Overview and general recommendations:

The submitted manuscript quantifies the contributions of soil nitrous oxide (N₂O) fluxes, and particularly from hot moments, to CO₂e balance for alfalfa agroecosystems. Using a novel dataset of multi-year, continuous, high-resolution chamber flux measurements coupled with belowground sensors and periodic soil sampling, the study highlights novel findings that hot moments of N₂O, although appearing in less than 2% of measurements, accounted for almost half of annual N₂O emissions and substantially offset the ecosystem carbon sink previously estimated for alfalfa. Consistent with recent literature, most hot moments occurred during soil re-wetting events, when anaerobic microsites may be generated, and were additionally modulated by temperature and by crop activities. Data collection, processing, and analytical methods are described well and are consistent with common practices of soil and ecosystem scientists.

After reading responses to reviewers and the revised manuscript, I am pleased with the changes the authors have made and I think revisions have made clear the importance and nuance of these results. I have a few comments regarding minor detail changes, but I think generally this manuscript is ready to move forward to publication. Great work!

We thank the reviewer for their comments.

Specific comments:

Abstract: I think somewhere in here you need to mention CO₂ and CH₄ as measured carbon fluxes since those are a big chunk of your results and you spend a lot of text describing them. The least invasive way to include them might be in Ln 18-19: "...offsetting the ecosystem carbon (CO₂ and CH₄) sink by ...". Or something more elegant. I think it's important to specify somehow that you looked at CH₄ in addition to CO₂, since readers may assume only CO₂ was measured here.

We agree, we have added "(carbon dioxide (CO₂) and methane(CH₄))" to L19.

Ln 38-39: Citations needed.

We removed this sentence as it is repetitive of third sentence of the paragraph, which is cited.

Ln 59-60: This citation does not have a matching reference in References section and is different annotation than other citations.

This has been corrected.

Ln 163-165: I'm getting confused by the wording here; can you write this sentence more simply? Is it that lower GPP tended to match low background N₂O?

The text (L171-176) has been changed to: "Increases in background (low-level) N₂O emissions were positively correlated with periods of high gross primary productivity (GPP), measured with satellite observations of near infrared reflectance of vegetation(NIRv)¹."

Reviewer #2 (Remarks to the Author):

Thank you for addressing many of my comments and clarifying certain issues. I still have major concerns as to how the results are interpreted and conveyed to readers. The main take away message is that 'alfalfa agriculture' emits significant amounts of N₂O, enough to offset the C-sink potential. The high overall annual N₂O emissions (=4 kg N/ha/yr with range of 1.6 to 5.7) occur over somewhat frequent and short periods of time associated with irrigation and driven by high N levels released by the high SOM content at the measurement site (Hot moments). I commented on the high SOM in my previous review and although the reply was that the site is on highly degraded peatland soil, the SOM content is still quite high compared to regular mineral soils. This means that the results obtained here can not be easily generalized to 'alfalfa agriculture' as the authors have done in Abstract and Conclusions. A much more nuanced interpretation of the measurements is needed. In fact, N₂O emissions are notoriously variable with soil type, management and weather events and generalizing results from one site is not possible. For example, Tenuta et al. (2019) using a micromet approach show that including perennial crops such as alfalfa in crop rotations leads to much lower N₂O emissions than annual crops, in contrast to this study.

We thank the reviewer for their thorough review and careful reading of the paper. We now better understand the reviewers concerns regarding SOM stocks described above and the need to include a more nuanced interpretation of our results. We have included additional text regarding this in L139-142.

With regard to generalizing these results to continuous alfalfa agriculture, we note that apart from this study, combined multi-year continuous flux measurements of CO₂, CH₄, and N₂O in continuous flood-irrigated alfalfa are essentially non-existent, even though this is the predominant practice for alfalfa in regions like the Western United States² (Text added in L93-96). Understanding the drivers of interannual variability, and accurately quantifying differences in emissions with stand age³ are needed to upscale emissions, particularly for N₂O which we show is inherently variable. To address the reviewers concerns we clarify that continuous measurements are needed to assess greenhouse gas emissions and the net C balance of continuous alfalfa ecosystems, as these are likely to differ from other agricultural activities including those that incorporate alfalfa in short-term rotations (L37-40).

The reviewer cites Tenuta et al. (2019) as an example of alfalfa agriculture leading to lower N₂O emissions than annual cropping. As mentioned previously, our goal here was not to compare alfalfa with annual cropping but to quantify continuous fluxes from multi-year continuous alfalfa agriculture, an important crop globally. This is fundamentally different from the mixed cropping system described in Tenuta et al. 2019. Regardless we have added some text to qualify our results stating, “This suggests net N₂O emissions from irrigated alfalfa may not always be reduced relative to other agricultural ecosystems receiving inorganic N inputs, particularly on relatively C-rich soils”. (L101-103).

In addition, the authors argue the large N₂O emissions measured in this study are due to the measurement method used. For example in L. 89-91 “Annual mean N₂O fluxes were 624.4 ± 26.8 mg and were significantly greater than other N₂O flux estimates in alfalfa using less intensive periodic sampling 30–33.”. While I strongly agree that more continuous measurements as provided by micromet methods are sorely needed and provide a much better flux time series, I think a more nuanced interpretation is also needed here. Firstly, the citations given here are not appropriate to support the argument made.

Ref 30: reports on emissions that are of a similar order of magnitude (2.3 and 5.7 kg N/ha/yr) with measurements made using static chambers;

Ref 32: is a micromet study with frequent observations;

Ref 33: reports on canola and wheat N₂O emissions following alfalfa termination for a semi-arid region.

We have changed and added nuance to the corresponding text (L92-96) and have added the following to the conclusions (L255-258): “This combination of automated chambers, eddy covariance, soil sensing, and satellite imagery is the most comprehensive dataset of multi-year annual budgets from continuous alfalfa agriculture to date, allowing us to determine the importance of both hot and non-hot moment emissions on total N₂O budgets and explore scale-emergent drivers of N₂O emissions.”

Long-term (> 2 years) continuous flux measurements, specifically of continuous alfalfa, are essentially non-existent. However, the above references, with the addition of Tanuta et al. 2019⁴ (although a grass/alfalfa mixture), and two additional references⁵, were some of the only potentially comparable annual estimates. We have removed the Malhi 2010⁶ reference and have clarified that some of these estimates are not from continuous alfalfa ecosystems (L94-96).

Secondly, as demonstrated by Ref30 the high emission peaks could be captured with other methods in an irrigated alfalfa system since they are quite predictable and timed with the irrigation. In fact, there may be an issue with overestimation given that chamber studies will typically target high emission events and then interpolate between these weekly data points to obtain annual emissions.

We agree that capturing high emissions events are important for calculating total budgets, and that our data show that irrigated fields may have more predictable hot moment fluxes

than other agroecosystems. In L115-119 we state that continuous measurements ensure all hot moments are captured as well as non-hot moment emissions that accounted for ~50% of the flux. We do not think that the chamber-based approach here was likely to overestimate fluxes because it was continuous. Micromet approaches cover a wider land area, but the comparatively high detection limit of most tower-based micromet N₂O measurement approaches (for example see Tenuta et al. 2016⁷, 2019⁴) could potentially lead to underestimated fluxes.

The final point of concern (which I missed in my first review) is the lack of consideration in C removal in harvest when determining if the field is a carbon sink. In fact, some information about forage yields and C content should be included.

We have added forage yield data provided by the farmer and cited for this field⁷ (L227-229). We agree that forage yields are helpful to account for total C fluxes from ecosystem-scale studies. However, we did not include the lifecycle of the alfalfa harvest once it left the field as the current research focuses on ecosystem-scale fluxes and not the full lifecycle of alfalfa products. We added this caveat to the manuscript (L230-233).

References

Tenuta, M., Amiro, B. D., Gao, X. P., Wagner-Riddle, C., Gervais, M. (2019). Agricultural management practices and environmental drivers of nitrous oxide emissions over a decade for an annual and an annual-perennial crop rotation. *Agricultural and Forest Meteorology*, 276, Article 107636. <https://doi.org/10.1016/j.agrformet.2019.107636>

Response References

1. Baldocchi, D. D. *et al.* Outgoing Near-Infrared Radiation From Vegetation Scales With Canopy Photosynthesis Across a Spectrum of Function, Structure, Physiological Capacity, and Weather. *Journal of Geophysical Research: Biogeosciences* 125, (2020).
2. Ottman, M. *et al.* Long term trends and the future of the alfalfa and forage industry. *Proceedings, 2013 Western Alfalfa & Forage Symposium, Reno, NV, 11-13, December, 2013. UC Cooperative Extension, Plant Sciences Department, University of California, Davis, CA 95616.* 11–13 (2013).
3. Burger, M., Haden, V. R., Chen, H., Six, J. & Horwath, W. R. Stand age affects emissions of N₂O in flood-irrigated alfalfa: a comparison of field measurements, DNDC model simulations and IPCC Tier 1 estimates. *Nutrient Cycling in Agroecosystems* 106, 335–345 (2016).
4. Tenuta, M., Amiro, B. D., Gao, X., Wagner-Riddle, C. & Gervais, M. Agricultural management practices and environmental drivers of nitrous oxide emissions over a decade for an annual and an annual-perennial crop rotation. *Agricultural and Forest Meteorology* 276–277, 107636 (2019).

5. Mosier, A. R., Halvorson, A. D., Peterson, G. A., Robertson, G. P. & Sherrod, L. Measurement of Net Global Warming Potential in Three Agroecosystems. *Nutr Cycl Agroecosyst* 72, 67–76 (2005).
6. Malhi, S. S., Lemke, R. & Schoenau, J. J. Influence of time and method of alfalfa stand termination on yield, seed quality, N uptake, soil properties and greenhouse gas emissions under different N fertility regimes. *Nutrient Cycling in Agroecosystems* 86, 17–38 (2010).
7. Hemes, K. S. *et al.* Assessing the carbon and climate benefit of restoring degraded agricultural peat soils to managed wetlands. *Agricultural and Forest Meteorology* 268, 202–214 (2019).